# Predicting chromatin conformation contact maps

**Alan Min**[1], **Jacob Schreiber**[2], **Anshul Kundaje**[2], **William S. Noble**[3,4]*

1 Department of Statistics, University of Washington, Seattle, Washington, United States of America,
2 Department of Genetics, Stanford University, Stanford, California, United States of America,
3 Department of Genome Sciences, University of Washington, Seattle, Washington, United States of America, 4 Paul G. Allen School of Computer Science and Engineering, University of Washington, Seattle, Washington, United States of America

* william-noble@uw.edu

## Abstract

Over the past 15 years, a variety of next-generation sequencing assays have been developed for measuring the 3D conformation of DNA in the nucleus. Each of these assays gives, for a particular cell or tissue type, a distinct picture of 3D chromatin architecture. Accordingly, making sense of the relationship between genome structure and function requires teasing apart two closely related questions: how does chromatin 3D structure change from one cell type to the next, and how do different measurements of that structure differ from one another, even when the two assays are carried out in the same cell type? In this work, we assemble a collection of chromatin 3D datasets—each represented as a 2D contact map—spanning multiple assay types and cell types. We then build a machine learning model that predicts missing contact maps in this collection. We use the model to systematically explore how genome 3D architecture changes, at the level of compartments, domains, and loops, between cell type and between assay types.

**Data availability statement:** All data used in this work is publicly available at the 4D Nucleome Data Portal, with accession codes listed in S2 Table. The Sphinx Python code is

## 1 Introduction

The spatial conformation of the genome inside a cell ("chromatin architecture") plays a key role in controlling cell type-specific cellular processes, such as regulation of gene expression and control of replication timing. An important aspect of this control involves bringing distal regulatory elements close, in 3D space, to the genes that they regulate. Consequently, aberrant changes to chromatin architecture can lead to diseases as genes become improperly regulated [1–4]. Hence, understanding endogenous chromatin architecture in each cell type and tissue in the body is crucial for understanding which alterations can disrupt these programs and what their anticipated effects are.

Accordingly, a variety of technologies have been developed to measure genome 3D conformation in a high-throughput fashion, allowing scientists to more fully explore the relationship between genome structure and function (Table 1). In the most commonly used genome-wide 3D conformation assay, dilution Hi-C, DNA-DNA contacts are counted by first fixing DNA using a cross-linker, thereby linking genomic regions that are close together [5]. The DNA is then cut using an endonuclease, and fragments of cross-linked DNA are ligated and

available, with an Apache license, at
https://github.com/Noble-Lab/Sphinx.

**Funding:** This study was supported by the National Institutes of Health in the form of grants awarded to WSN (UM1HG011531 and R01HG011531) in the form of salary for WSN and AM. The specific roles of these authors are articulated in the "author contributions" section. The funders had no role in study design, data collection and analysis, decision to publish, or preparation of the manuscript. No additional external funding was received for this study.

**Competing interests:** The authors have declared that no competing interests exist.

**Table 1. Properties of assays measuring 3D genome architecture generated by 4DN and included in this study.**

| Assay | Year | Targets a subset of contacts | Cleaves using endonuclease | Barcoding based | Improved resolution |
|---|---|---|---|---|---|
| Dilution Hi-C [5] | 2009 | | ✓ | | |
| ChIA-PET [6] | 2009 | ✓ | ✓ | | |
| in-situ Hi-C [7] | 2014 | | ✓ | | ✓ |
| Micro-C [8] | 2015 | | ✓ | | ✓ |
| PLAC-seq [9] | 2016 | ✓ | ✓ | | |
| DNase Hi-C [10] | 2016 | | ✓ | | ✓ |
| DNA SPRITE [11] | 2018 | | | ✓ | |

sequenced to produce counts. A variety of modifications to this general protocol have been proposed subsequently. For example, in situ Hi-C [7], micro-C [8], and DNAse Hi-C [10] aim to improve resolution and signal-to-noise ratio of Hi-C based assays by either performing ligation in the nucleus (in-situ Hi-C) or using a higher resolution nuclease (micro-C or DNAse Hi-C). ChIA-PET [6], PLAC-seq [9] and Hi-Chip [12] use immunoprecipitation to target DNA-DNA contacts that are mediated by a protein of interest. In contrast, SPRITE [11] and GAM [13] do not rely on ligation. SPRITE uses "split pooling," in which cross-linked molecules are tagged with the same barcodes to identify neighboring genomic regions. GAM uses cryosectioning in conjunction with sequencing and identifies genomic regions as neighboring if they are frequently observed in the same cryosection regions. The output of each such assay can be summarized in a *contact map*, a matrix containing counts of observed contacts between pairs of genomic loci.

The National Institutes of Health 4D Nucleome Network (4DN) [14,15] has collected chromosome conformation experiments from multiple labs involving many cell or tissue types (generically, "biosamples") and assays. However, it is not feasible to carry out an experiment of every type of assay in every biosample, because chromosome conformation experiments are complex and costly and there are many biosamples that one may wish to interrogate. Therefore, we decided to pursue the hypothesis that machine learning methods can provide draft characterization of chromosome conformation across a wide range of biosamples and assays for which experimental data is not yet available, using only experiments that have already been performed. We can summarize the available experiments in a 2D matrix, with eight rows corresponding to different 3D chromatin assays and 11 columns corresponding to different biosamples (Fig 1). Each entry in the matrix entry is a contact map, and among the 88 possible contact maps, 4DN has carried out 41. Our goal is to accurately impute the missing 47 contact maps.

Such predicted contact maps have a variety of possible uses. A systematic analysis of a complete collection of contact maps may provide global insights into how important aspects of chromatin architecture, e.g. TAD boundaries and loops, differ or are similar across biosamples and assays. In some cases, it may not be possible to run multiple 3D chromatin assays on a particular sample due to the material necessary for each experiment. Interesting features in a predicted contact map may suggest hypotheses for follow-up experiments or may help in prioritization of future 3D chromatin assays. Conversely, if the predicted contact map for a given biosample or assay closely resembles one or more existing contact maps, that suggests that conducting the experiment may not be worthwhile. In these cases, predicted contact maps for other assay types may give an idea of what the results would have been if they had been done.

Several methods have been proposed previously to predict Hi-C contact maps, or portions thereof. Some methods, such as Akita [16], DeepC [17] and Orca [18], take as input the DNA

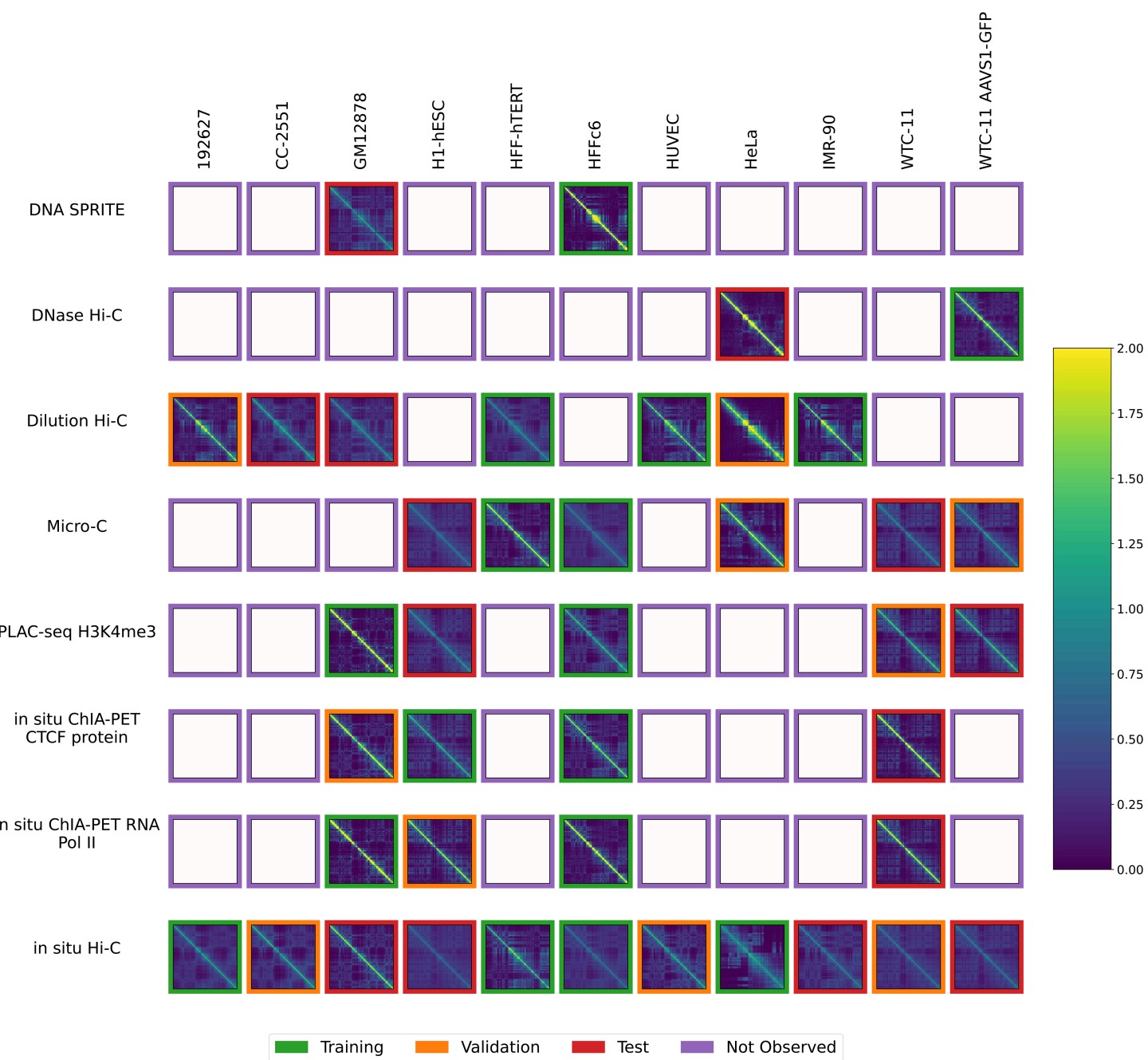

**Fig 1. Available contact maps from 4DN.** Each panel displays a contact map for chromosome 19 from a particular assay and a particular biosample where the number of normalized log counts is displayed as a color. Only cell types with at least two experiments and assays with at least two experiments were included. Each of the 41 non-missing contact maps has a colored border to indicate whether it was used for training (green), validation (orange) or testing (red).

sequence and produce as output a corresponding contact map. Akita and DeepC predict only contacts close to the diagonal of the matrix, whereas Orca can make predictions for an entire chromosome at a time. However, these sequence-only models can only make predictions for the set of experiments that they were trained on and so cannot generalize to cell types or tissues without experimental data. Alternatively, HiC-Reg [19] uses a random forest regression

model to predict a Hi-C contact counts based on epigenomic features of the pair of anchoring genomic intervals. Similarly, Epiphany [20] predicts Hi-C contacts from 1D epigenomic data, including ChIP-seq measurements of chromatin accessibility, CTCF binding, and several types of histone modifications. Alternatively, DeepChIA-PET [21] predicts ChIA-PET contacts from a combination of Hi-C and ChIP-seq data. Despite being able to generalize to biosamples that they were not trained on, these methods are limited to making predictions for a single type of contact map and require that a fixed set of epigenomic experiments have been performed.

In contrast to these approaches, we consider the complementary problem of imputing unseen contact maps given only a collection of available contact maps. This imputation task is conceptually similar to previous work on imputing 1D epigenomic data [22–24]. In that setting, large compendia of 1D epigenomics experiments were organized into a three dimensional tensor with the axes being biosample, assay, and genomic position, and computational models learned from the structure of the available measurements to make predictions of unseen experiments. In our setting, our data is organized into a four dimensional tensor with the additional axis being the second genomic position, as contact maps are comprised of interactions between pairs of positions.

We introduce Sphinx, a deep tensor factorization method for imputing entire contact maps. Sphinx is comprised of learned representations of biosamples, assays, and genomic positions, and a neural network that combines these factors to make predictions. We evaluate Sphinx in a cross-validated setting, demonstrating that it outperforms several baseline methods. We then systematically apply Sphinx to the full 4DN dataset in a prospective fashion, and we examine the predicted contact matrices on three levels: contact decay profiles, compartments, and topologically associating domains (TADs). The results suggest an increased capability to compare contact matrices, and a smoothing effect on similarity between biosamples and assays.

## 2 Methods

### 2.1 Sphinx

The deep matrix factorization approach was implemented in a similar fashion to the Avocado imputation approach [24]. We used Pytorch version 2.0.1 with Python 3.10.12 for the implementation. We created embedding layers for cell type, assay, and genomic position. We then select two genomic position factors associated with the two positions required along with the cell type and assay factors. We further include a set of distance factors encoding the distance between the two genomic positions. These are concatenated and used as input to a series of fully connected layers with equal number of hidden nodes. The final output layer is a single float value. We constrain the first position factor to be less than the second to enforce symmetry of the prediction. These predictions are then reflected across the diagonal. A schematic of the architecture of the Sphinx model is shown in Fig 2. We conducted a hyperparameter search using a partial grid search spanning initial learning rate, dropout, and the number of cell type factors, assay factors, position factors, layers, and nodes (Fig 3a). The Adam optimizer [25] was used to optimize our model. We randomly chose 500 of 9126 hyperparameters to train models, and each model was trained up to at most 50 epochs using a batch size of 10,000 examples, each example being one matrix entry of one experiment. Each epoch consisted of 526 batches. Batches were generated from randomly selected off-diagonal elements of each of the training experiments. Validation error was calculated at the end of each epoch, and model selection was based off the best validation error. The Sphinx Python code is available, with an Apache license, at https://github.com/Noble-Lab/Sphinx.

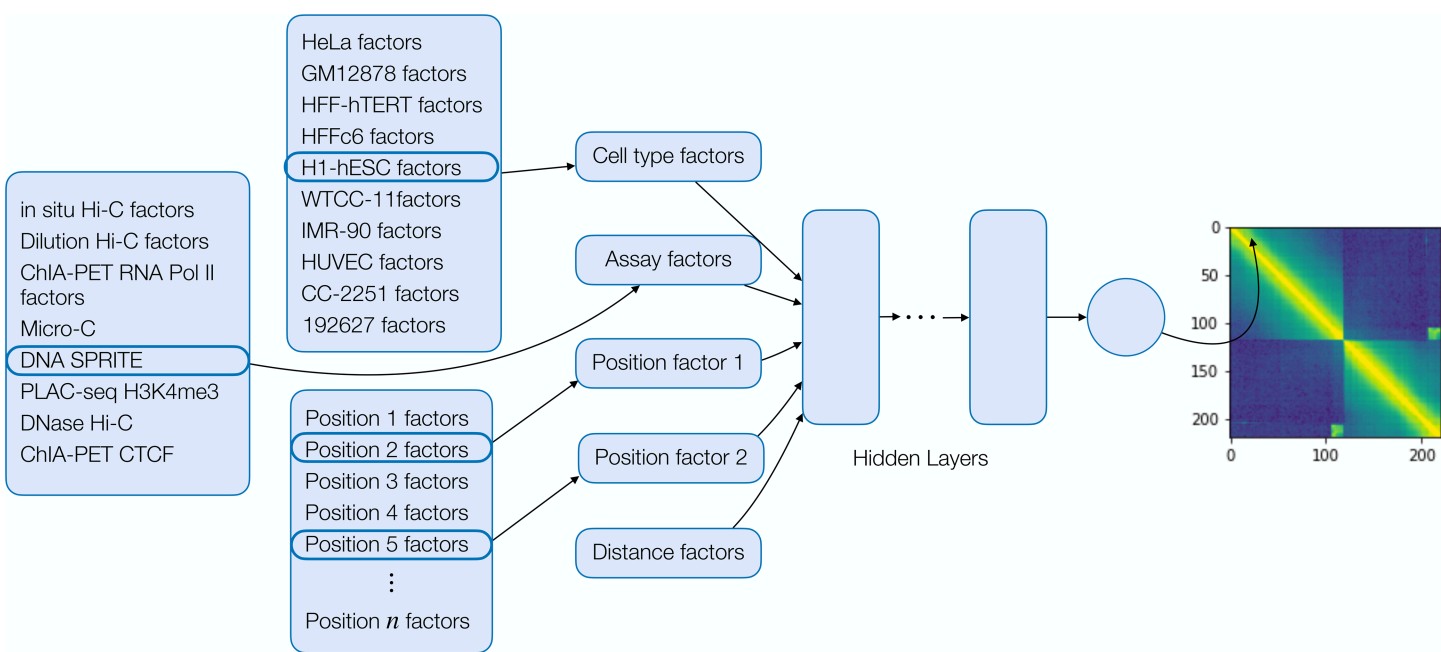

**Fig 2. The Sphinx model.** Each dimension of the 4D input is encoded using latent factors, and the concatenated factors are processed by a multi-layer perceptron. The figure shows an example of predicting the contact between positions 2 and 5 in DNA SPRITE H1-hESC. The final output of the model is the predicted normalized contact count.

## 2.2 Data sets

We systematically downloaded all available contact map datasets from the 4D Nucleome web portal (https://data.4dnucleome.org/) and assembled them into a biosample-by-assay type matrix. Rows or columns with a single entry were iteratively deleted until every row and column contained at least two non-missing entries. The final data set contains eight assay types (S1 Table), 11 biosamples (Table 2), and 44 contact maps (Fig 1 and S2 Table). We primarily trained the model on one chromosome (chromosome 19 for most analyses, and chromosome 1 in Sect 3.4) at 100 kb resolution. We also experimented with training our model at 10 kb resolution but were unable to improve upon the baseline method (S1 Sect).

Table 2. Biosamples included in the data set

| Biosample | Description |
| --- | --- |
| HeLa | An immortal cell line derived from cervical cancer cells from Henrietta Lacks |
| GM12878 | Lymphoblastoid cell line from a woman of European descent |
| HFFc6 | Human foreskin fibroblast cells |
| hESC | Human embryonic stem cells |
| WTC-11 | Pluripotent stem cells derived from leg fibroblast |
| IMR-90 | Lung fibroblast cells |
| HUVEC | Human umbilical vein endothelial cells |
| CC-2251 | Mammary epithelial cells |
| 192627 | Adult normal human epidermal keratinocytes |

Prior to analysis, each contact map is normalized as follows. First, we $\log(x + 1)$ transform each contact map entry to avoid high dynamic range of the contact counts and to avoid taking the log of 0. Second, we discard rows and columns of each chromosome matrix with low marginal counts. This is done by creating a separate vector of marginal counts for each contact map. We then discard any bin that is beyond 10 mean absolute deviations from the median marginal count in any one of the contact maps in the training set. In practice, this procedure eliminates 31 of the 587 of the bins resulting in 556 pruned bins. To avoid leakage, the same set of bins (identified using the training data) is also eliminated from the validation and test sets. Third, we normalize the sum of the remaining $\log(x + 1)$ transformed contacts to sum to $10^5$ to avoid detecting differences in read depth.

The data was randomly partitioned into train, test and validation sets as illustrated in Fig 1. For each assay, we randomly chose experiments of a single data type one by one and assigned them to the training, test, and validation set in that order. This ensured that when assays only had 2 examples, the training and test set each had one example of that assay. This resulted in 17 training experiments, 10 validation experiments, and 14 test experiments.

## 2.3 Baseline methods

Due to the key differences between Sphinx and existing methods described in Sect 1, there are no appropriate alternate methods to compare with Sphinx. Without any other predictive models to directly compare to, we consider three baseline methods: the row mean, the column mean, and the cross mean. In previous imputation studies, this type of mean predictor provides a strong baseline [24,26]. The row mean is the average of all observed data that shares the same assay as the queried matrix. The column mean is the average of all observed data that shares the same biosample. The cross mean takes the average of all data that share either biosample or assay with the queried matrix.

## 2.4 Computing the contact decay, eigenvector, and insulation score

To compute the contact decay profile, we computed the we computed the mean normalized count along each of the diagonals of the contact matrix. To compute the eigenvector, we used NumPy version 1.25.2. To compute the insulation score, we used a window size of 30, and computed the mean of sliding window diagonal submatrices of the contact matrix, sliding the window by 1 bin at each step.

# 3 Results

## 3.1 Sphinx accurately imputes unobserved contact maps

To validate our modeling approach, we began by measuring how well the model predicts previously unseen data. To this end, we randomly selected a subset of contacts map to use for validation and testing (Sect 2.2). As our primary performance measure, we used the mean squared error (MSE) between the observed and imputed contact maps, which corresponds to the loss function optimized by Sphinx. Because, to our knowledge, this type of imputation has not been previously reported in the literature, we designed several baseline methods against which to compare the performance of Sphinx. These involve averaging over rows, columns, or both (i.e., averaging in a cross-shaped pattern centered on the target matrix) (Sect 2.3).

Our experiment began by using the validation set to select hyperparameters for Sphinx. In this step, we defined a seven-dimensional grid of hyperparameters, including learning rate, cell type factors, assay factors, position factors, the number of layers, the number of nodes per layer, and the dropout rate. Because this grid is too large to explore completely, we opted

to randomly select 500 points on the hyperparameter grid and use those as the basis for our selection.

The results of this hyperparameter search suggest that Sphinx successfully outperforms the baseline methods with a variety of hyperparameter settings (Fig 3a). Among the 500 possible hyperparameter settings, Sphinx's validation MSE is less than the cross-mean MSE in 306 cases. The best-performing hyperparameter combination achieves an MSE of 0.052, which is 6.32% lower than the MSE achieved by the best-performing baseline (MSE of 0.056 from the cross-mean model).

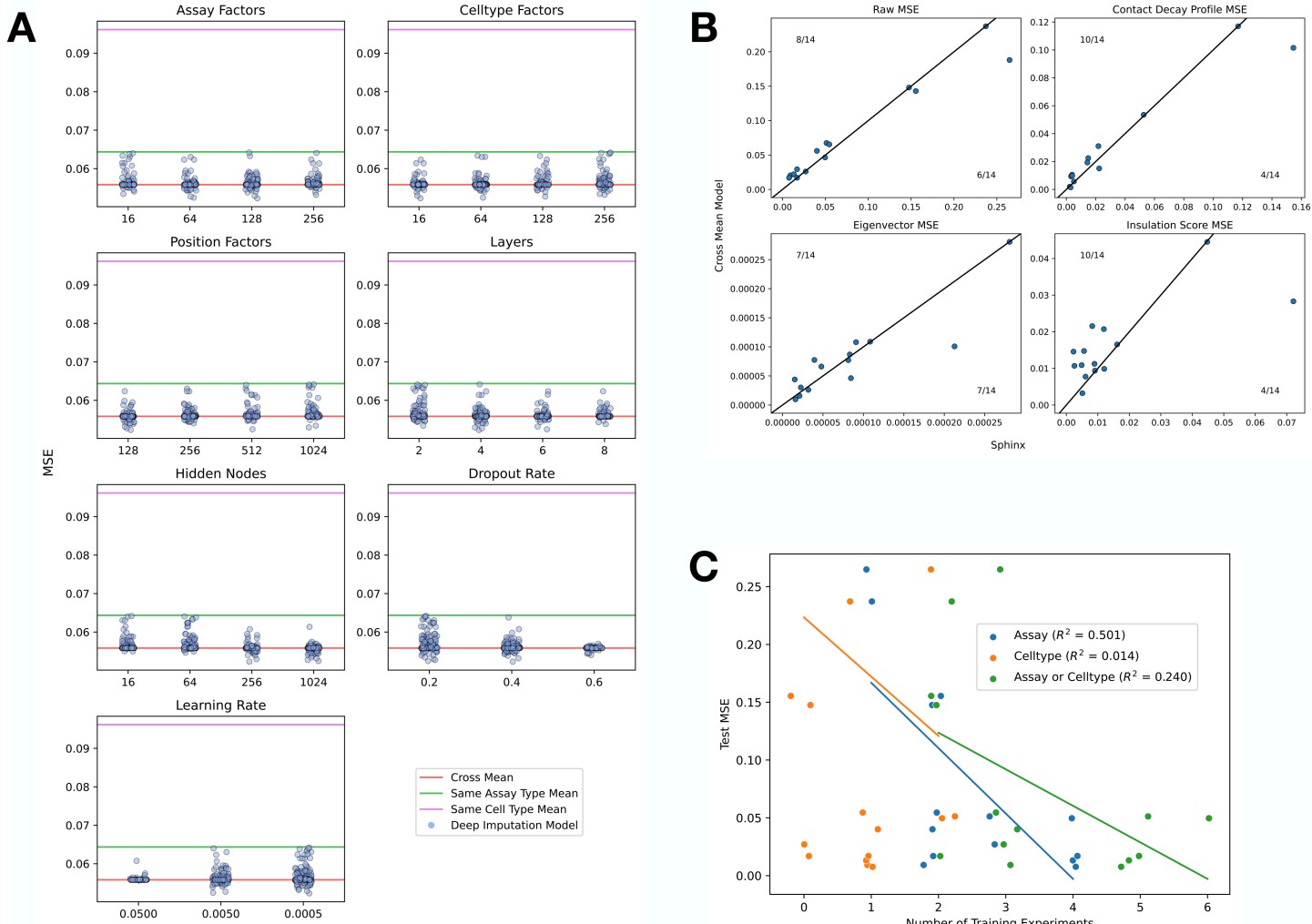

**Fig 3. Sphinx outperforms mean model baseline**. (a) Sphinx was trained using 500 randomly selected settings of seven different hyperparameters. Each panel plots the validation set MSE (x-axis) for various values of a particular hyperparameter (y-axis). MSEs are shown as horizontal lines for the three baseline methods: cross-mean (red line), same cell type (magenta), and same assay type (green). All analyses are based on data from chromosome 19. (b) Each panel plots the test set MSE achieved by Sphinx (x-axis) versus the cross-mean baseline (y-axis). Each point corresponds to one of the 14 assay/cell type combinations in the test set. (c) A comparison of the number of training experiments that shared assay (blue), cell type (orange), or either assay or cell type (green) versus the MSE of the test set example. Each point is one test set example. The x-axis is jittered for visibility. Least squares lines are shown along with their associated $R^2$ values.

Next we evaluated how well the selected hyperparameters generalize to the test data. This analysis suggests that Sphinx performs comparably to the best-performing baseline (cross-mean). In particular, computing the overall MSE on the test set, we find that Sphinx outperforms the baseline in 8 out of 14 cases (Fig 3b). We also compared the MSE of Sphinx depending on the number of training set examples that shared either the assay, celltype, or either assay or cell type with the test set example (Fig 3c). We observed that the greatest association with decrease in test MSE is with increased number of training set examples that have the same assay as the test set ($R^2$ = 0.501). We also confirmed that removing training examples increased the validation MSE (S2 Fig). We also computed the Pearson correlation as a function of genomic distance (S4 Fig)

## 3.2 Imputation model reveals similarities between biosamples

Having established that Sphinx produces comparable predictions to the cross-mean baseline, we next set out to systematically apply Sphinx to the full dataset. The goal here is to provide a data resource that enables exhaustive comparison of all cell lines and all assay types. Without imputation, many comparisons would be incomplete. For example, based on the pattern of missingness in Fig 1, if we want to compute the similarity between the GM12878 and HeLa cell lines, we could do so only on the basis of dilution Hi-C and *in situ* Hi-C data. In particular, this comparison would not make use of six data sets: HeLa data from DNase Hi-C and dilution Hi-C and GM12878 data from DNA Sprite, PLAC-seq, and two ChIA-PET experiments. Furthermore, some comparisons would be impossible to compute: in our dataset, three pairs of assay types have not been carried out in any common cell lines: DNA SPRITE and DNase Hi-C as well as DNAse Hi-C and both ChIA-PET assays.

Accordingly, we used the previously trained model to impute contact maps for all unobserved assay-cell type combinations (Fig 1). Also, for each imputed contact map, we computed three types of features: contact decay profiles, eigenvectors, and insulation scores (Figs 4–6). For comparison, we also carried out the imputation and feature calculation using the cross-mean model. In each case, we see good qualitative agreement between the observed and imputed values in the test and validation sets.

Finally, based on these imputed values, we computed complete pairwise correlation matrices for cell types and assay types (Fig 7). To obtain these correlations, we first calculated the contact decay profile, eigenvectors, or insulation score profile of each of the imputed matrices (Sect 2.4). Next, we selected two cell types (or assay types), $c_1$ and $c_2$. We then computed the mean Pearson correlation between the experiments $(c_1, a)$ and $(c_2, a)$ for each of these data modalities, where $a$ is a shared assay (or cell type). We also computed these mean correlation values using the unimputed data, skipping any experiments that were not available. The correlation matrix based on imputed values is complete and thus allowed us to carry out hierarchical clustering of cell types and assay types, which is not possible using the unimputed data due to the presence of missing values. We noted that in the unimputed correlation matrix, the entries with the greatest dissimilarities are often based on only a few experiments. When using the imputed matrices for the correlation matrix, these values often became less extreme. For example, in the contact decay profile assay similarities, dilution Hi-C appears to be dissimilar from PLAC-seq and DNase Hi-C (unimputed Pearson correlations of 0.89 and 0.86, respectively, compared to imputed values of 0.96 and 0.97), and DNase Hi-C is dissimilar from *in situ* Hi-C (unimputed Pearson correlation of 0.83 compared to 0.95). However, there is only one matrix with shared cell type between dilution Hi-C and PLAC-seq, and likewise with DNase Hi-C. In the case of DNase and in situ Hi-C, only two experiments

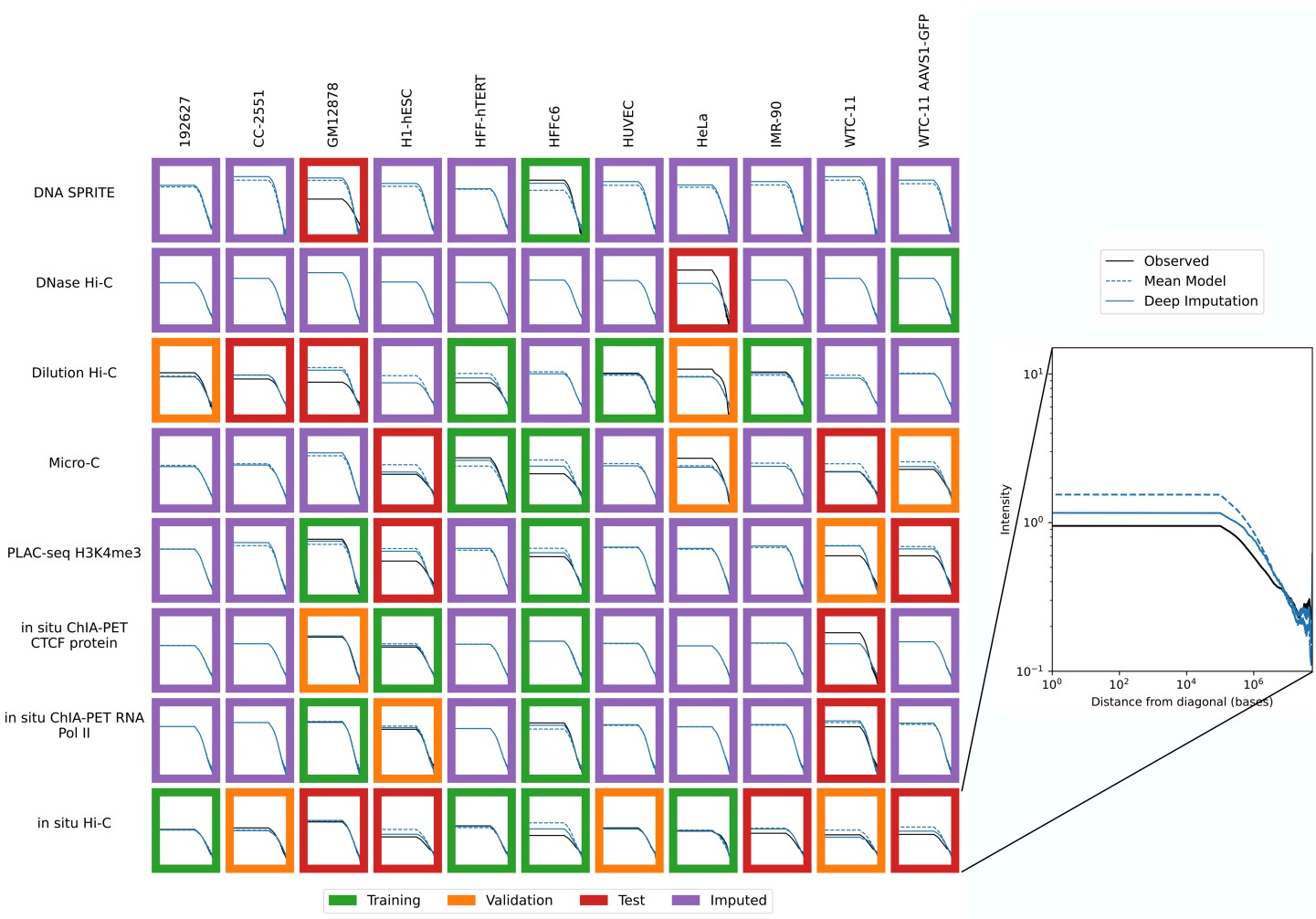

**Fig 4. Imputation enables visualization of unobserved contact decay profiles.** Contact decay profiles for training (green outline), validation (orange outline), test (red outline) and unobserved data (purple outline) are shown. The inset shows an example of one contact decay profile with the distance from the diagonal (X-axis) plotted against the intensity (Y-axis), which is the mean value for all normalized contacts at that distance from the diagonal. Axes are consistent across all panels.

are shared. This sparsity makes these computed similarities potentially less reliable. In practice, we observed that in these cases, the imputed correlations are higher than the original correlations.

## 3.3 Imputation reveals hypotheses for contact decay profiles and insulation score

Using our held out validation and test datasets, we can verify that hypotheses generated by our imputations are plausible. For example, when we consider contact decay profiles (Fig 4), we observe that for micro-C in H1-hESC, which is a test set example, the Sphinx imputed contacts near the diagonal are less prominent than the mean-model imputed contacts near the diagonal. Previous work has found that embryonic stem cells tend to exhibit a greater number of long-distance contacts than more differentiated cell types [27]. In our test example, Sphinx accurately predicts this phenomenon. Sphinx also accurately imputes less frequent short-range contacts for micro-C in the WTC-11 cell line, which is an induced pluripotent

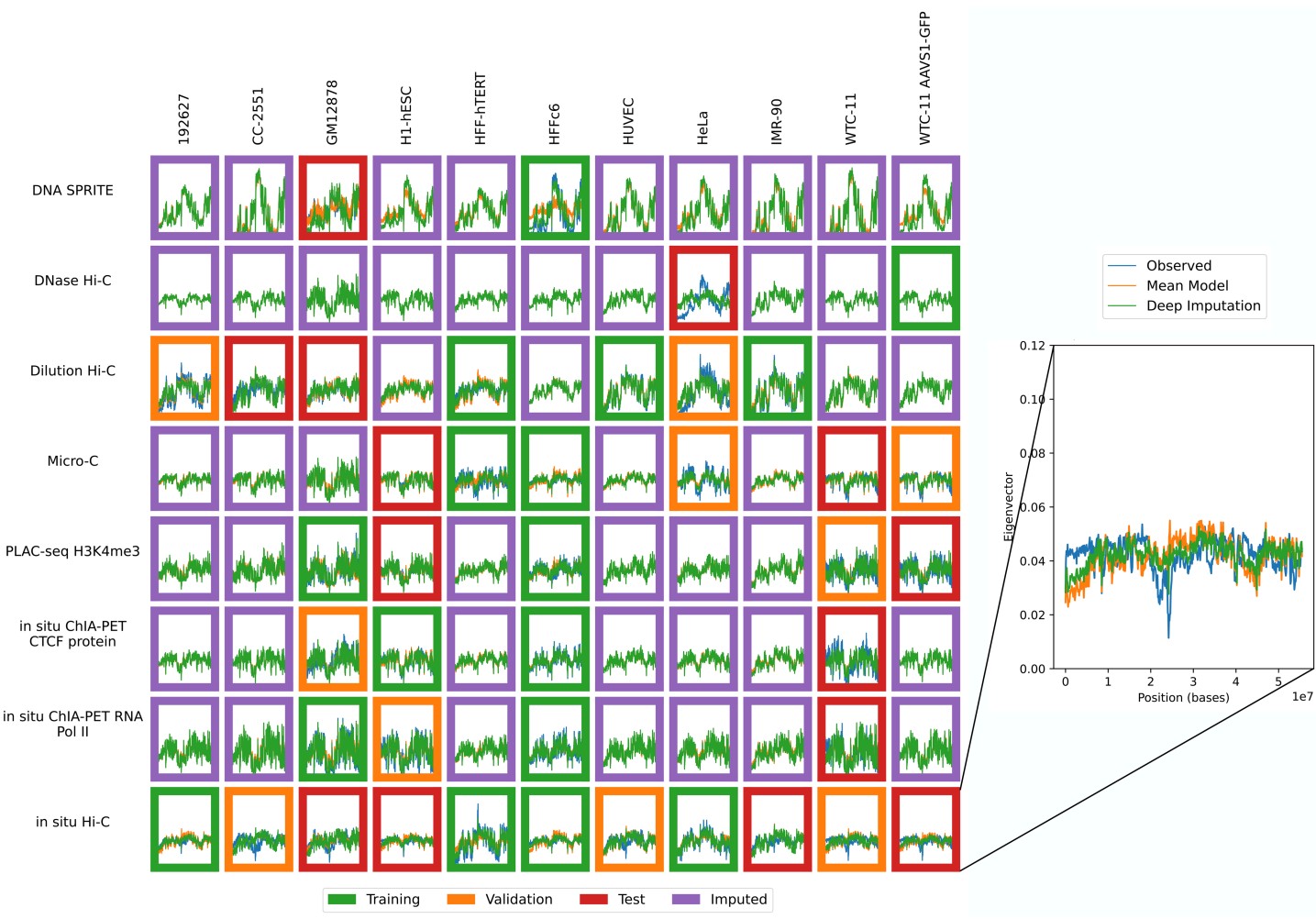

**Fig 5. Imputation enables visualization of unobserved eigenvector profiles.** Eigenvector profiles for training (green outline), validation (orange outline), test (red outline) and unobserved data (purple outline) are shown. The inset shows an example of the eigenvector profile with the position (X-axis) plotted against the eigenvector value (Y-axis). Axes are consistent across all panels.

stem cell line. These two observations suggest that Sphinx may be able to infer characteristics of less differentiated cell lines, which may otherwise be drowned out in the cross-mean average baseline method. Sphinx also predicts, for the unobserved micro-C experiment in GM12878, increased short-range interactions compared to the cross-mean baseline, which may be reflective of the fact that GM12878 is a more differentiated lymphoblast cell line, compared to the embryonic stem cells in either H1-hESC or WTC-11. Interestingly, Sphinx predicts that the unobserved micro-C experiment for IMR-90, which is a fibroblast cell line, has decreased close-range contacts compared to the baseline.

Another example can be seen in the insulation scores for dilution Hi-C (Fig 6). For one of the test set examples, GM12878, we observe several regions where Sphinx accurately predicts lower insulation scores than the mean model. Although in the CC-2551 example, Sphinx predicts a very similar profile to the mean model, these two examples suggest that insulation scores computed from Sphinx results are plausible. We observe that there are various regions

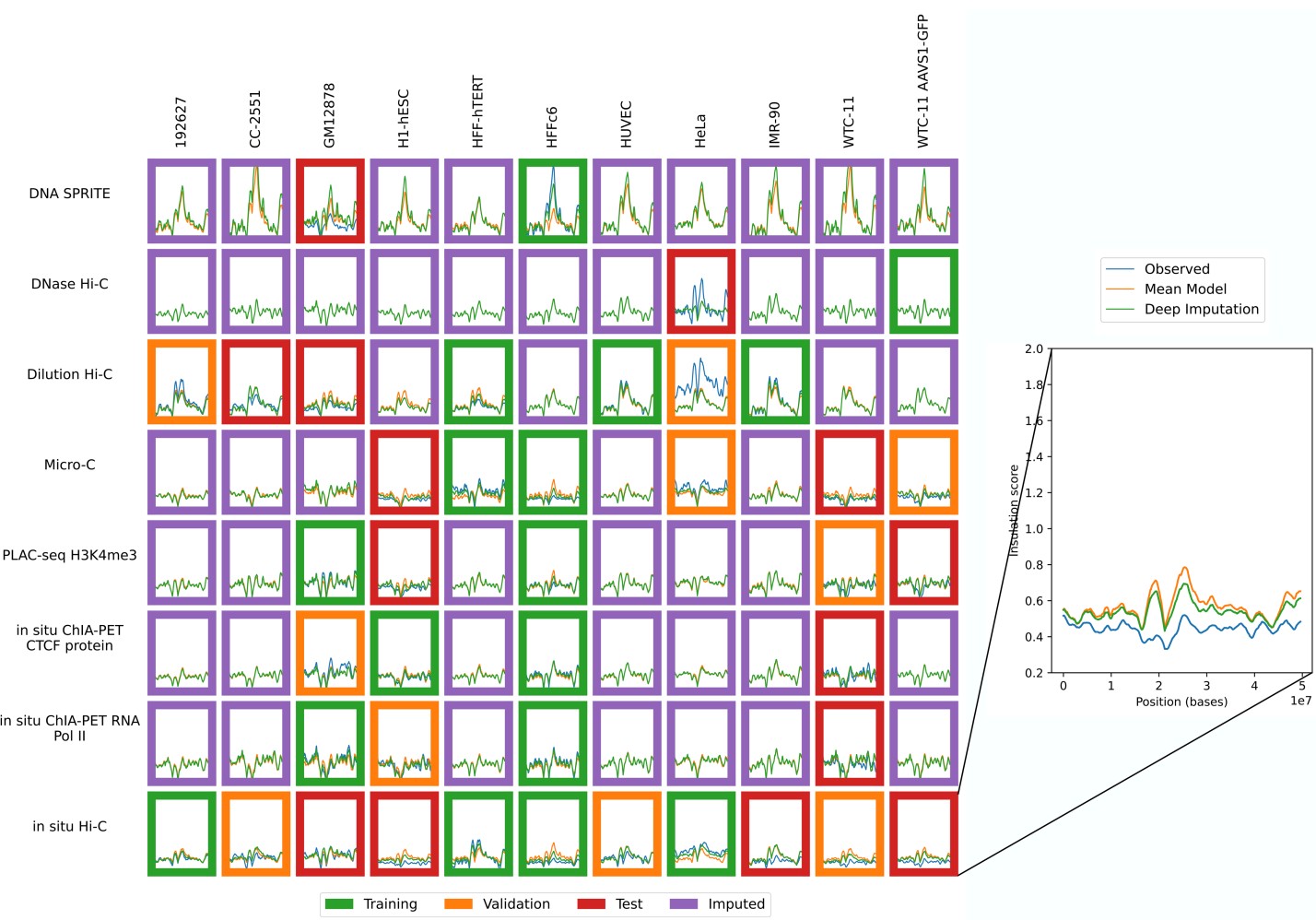

**Fig 6. Imputation enables visualization of unobserved insulation score profiles.** Insulation score profiles for training (green outline), validation (orange outline), test (red outline) and unobserved data (purple outline) are shown. The inset shows an example of the insulation score profile with the position (X-axis) plotted against the insulation (Y-axis). Axes are consistent across all panels.

in the H1-hESC cell type where Sphinx predicts lower insulation scores than the mean model, which may be an area that warrants further investigation.

### 3.4 Sphinx on larger scale data

To investigate how Sphinx's performance changes when trained from more data, we trained the model on chromosome 1 with the same set-up as when trained on chromosome 19. We kept the sum of the normalized logged counts the be same as in chromosome 19, which meant that on average, the counts when we trained chromosome 1 were lower because there are more bins in chromosome 1. Chromosome 1, after pruning, was $2232 \times 2232$, compared to $556 \times 556$ in chromosome 19. We reasoned that because there were changes in sparsity and number of bins, we should conduct a hyperparameter search for chromosome 1. Because there were more bins in chromosome 1, training this model was more computationally intensive, and hence we were unable to conduct a full hyperparameter search. We checked around 150 hyperparameter combinations, as opposed to the 500 combinations in chromosome 19.

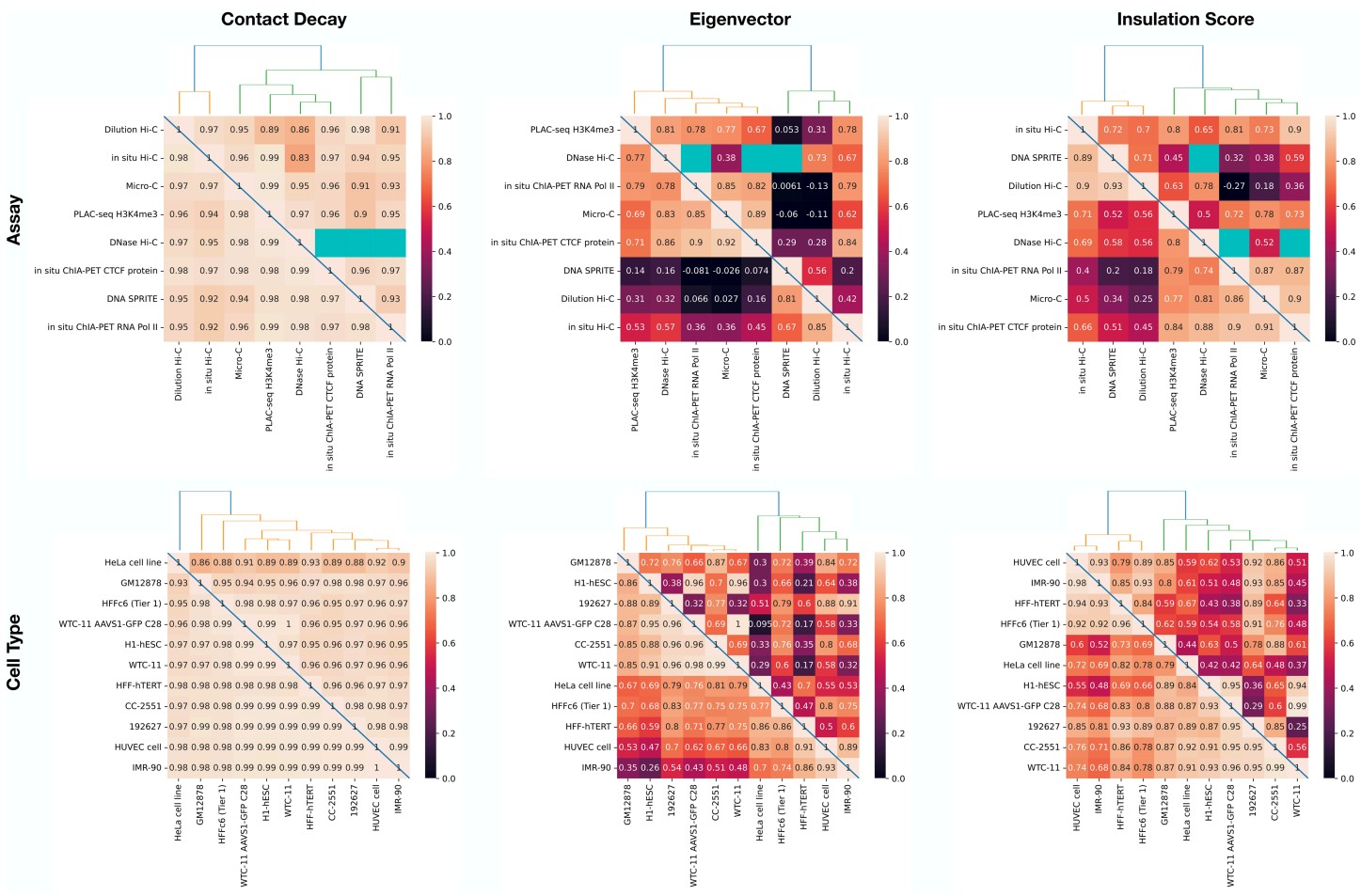

**Fig 7. Comparisons of cell types and assay types based on contact decay, eigenvectors, and insulation scores**. In each panel, the Pearson correlation is calculated between the (left column) contact decay profiles, (middle column) eigenvectors, and (right column) insulation scores for each of the cell types with shared assays (top row) and assays with shared cell types (bottom row). The lower triangle is the correlation after missing contact maps have been imputed, and the upper triangle is correlation with no imputation. Blue squares indicate comparisons that were not possible due to unobserved experiments. The dendrogram is computed using the imputed values.

The deep imputation model outperformed the cross-mean baseline model in validation in 89 hyperparameter combinations, but visually, the most significant improvements were made when the initial learning rate was 0.0005, which was not observed in chromosome 19 (Fig 8A). This difference may be because the average bin had lower counts in chromosome 1 due to our normalization scheme. Unfortunately, we found that in the test set, Sphinx performed similarly or slightly worse than the baseline model using our metrics of the raw MSE, contact decay profile MSE, eigenvector MSE, and insulation score MSE (Fig 8B). We confirmed visually that the model seemed to have converged (Fig 8C). We hypothesize that the decreased performance compared to chromosome 19 may be due to an increase in sparsity in chromosome 1 (Fig 8D), or because the hyperparameter search was not as deep.

We further attempted to run Sphinx at 10 kb on chromosome 19 (S1 Sect); however, we suspect that many of the same issues we experienced with chromosome 1 were exacerbated at 10kb resolution. Namely, there was increased sparsity at 10kb resolution and due to computational constraints, we did not conduct as extensive of a hyperparameter search.

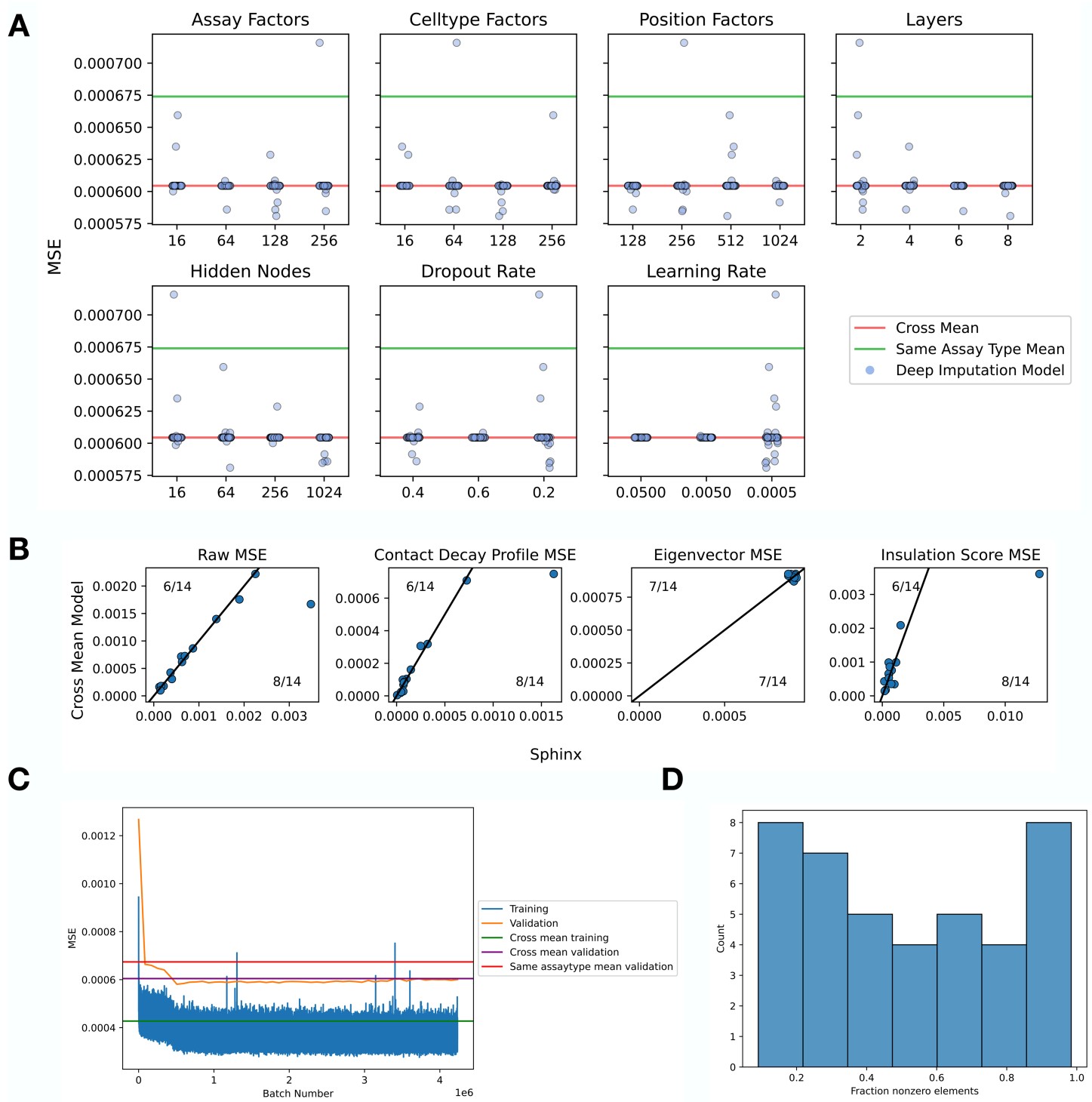

**Fig 8. On sparser data, Sphinx outperforms the baseline in validation data, but not in test data.** (A) We conducted a hyperparameter search using the standard Sphinx model for chromosome 1 with around 150 hyperparameters tested. (B) Raw MSE, contact decay profile MSE, eigenvector MSE, and insulation score MSE are compared between Sphinx and the cross-mean model, similarly to in Fig 3. (C) The training curve, validation curve, and baselines for the best performing model on chromosome 1. The training curve is a rolling average of 10 batches. (D) A histogram of the fraction of nonzero elements in all data samples in chromosome 1.

## 3.5 Experimenting with a convolutional architecture

A convolutional architecture (Fig 9A) was also implemented as an alternative to the fully connected layer. This architecture is similar to the main Sphinx model except that that neighboring positions were convolved with each other before being concatenated with the cell type and

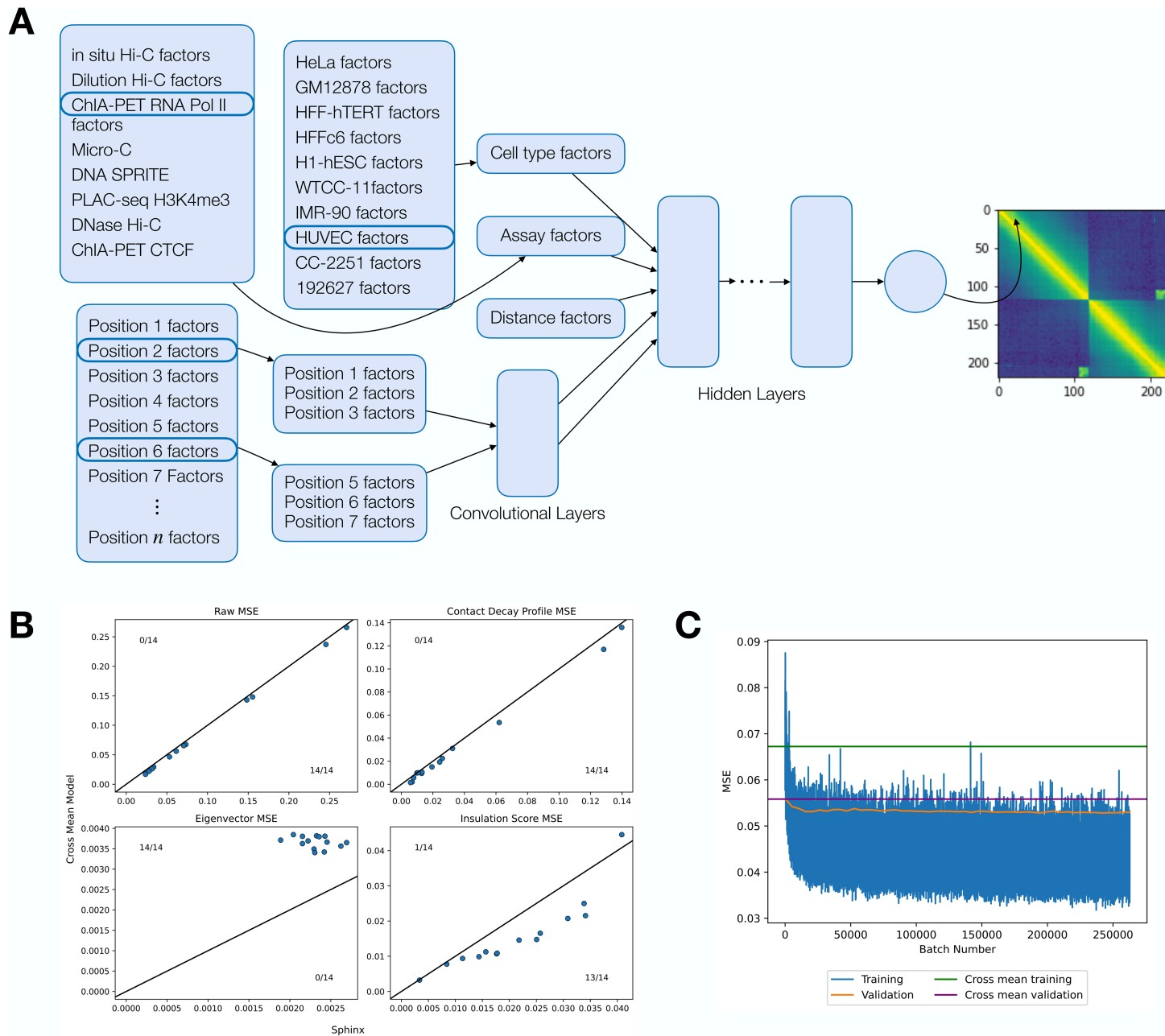

**Fig 9. Convolutional Sphinx architecture does not improve performance over standard architecture** (a) Cell type and assay factors are input into the hidden layers as in Fig 2. Position factors are first convoluted in a window of neighboring factors. The window size is shown as 1 in the figure as an example. The result of the convolution is input into the hidden layers. (b) A comparison of the raw, contact decay profile, eigenvector, and insulation score MSEs. The black line indicates *y = x*. (c) The training curve (rolling average over 10 batches) and validation curve (plotted per epoch) are shown and compared to baselines.

assay factors. If the position of interest was too close to the end of the chromosome, then we set the windowed positions outside of the chromosome to be 0. We trained the convolutional model using chromosome 19 at 100kb resolution using the best-performing hyperparameters for the base Sphinx model (16 celltype factors, 128 assay factors, 128 position factors, 128 distance factors, 256 hidden nodes, 4 hidden layers, with a dropout of 0.4, and an initial learning rate of 0.0001). We set a window of 10 positions before the central position and 10 positions after, for a total of 21 sets of position factors being entered into the convolutional neural network. We used two 2D convolutional layers. The first had one input channel, three output channels, and a kernel size of 21. The second had three input channels, one output channel, and a kernel size of one. The output of these two convolutional layers was input into the fully connected layers, and the rest of the model was the same as in the main Sphinx model. The model was trained for 50 epochs, and the iteration with the best validation error was kept.

We used the Sphinx model to make predictions on each of the test set examples, as in Fig 3, and we compared to the MSE from the mean model and the convolutional Sphinx model. We found that in terms of raw test-set MSE and contact decay profile MSE, the convolutional Sphinx model performed marginally worse than the mean model. The eigenvector MSE was better in the Sphinx model for all cases, and the insulation score was worse in most cases (Fig 9B). We determined that the average Sphinx validation MSE was overall improved upon the average mean model validation MSE and that the model appears to have converged upon visual inspection (Fig 9C). We noted that these results do not represent large improvements over the standard Sphinx model and hence did not continue to pursue the convolutional model.

## 4 Discussion

Each distinct 3D contact map assay provides unique insights into the biology of the cell type that it is used on. Some methods, such as in-situ Hi-C, micro-C, or DNase Hi-C, aim to provide global contact maps. Other methods, such as ChIA-PET or PLAC-seq, focus on interactions mediated by particular proteins. DNA SPRITE includes contacts that are physically more distant, such as in the case of nuclear bodies. These distinct purposes can provide complementary insights.

Although many chromatin 3D contact maps have been collected for combinations of assays and human cell types, there are still many combinations that have not been collected. This missingness motivates a computational method to impute unobserved experiments in order to generate hypotheses and enable downstream analyses that require complete observations. We first developed baseline methods that take the average of experiments either using only the same assay, the same cell type, or either. We then improved on these baseline methods using our machine learning model to predict areas where the baseline method was most likely to be incorrect. We demonstrated that we improved upon the baseline methods in many of the validation cases and more often than not in the test set (Fig 3A, 3B).

In our matrix of 88 potential contact maps from the 4DN web portal, we produced imputed experiments for the 47 unobserved experiments (Fig 10). We then conducted an analysis to determine similarities between assays and cell types across contact decay profile, eigenvector, and insulation score (Fig 7). We found that we were unable to accurately predict loops, however (S2 Sect). These analyses identify characteristics that are unique to individual cell types and assays. We demonstrated that without the imputed data, some comparisons cannot be made due to insufficient data. Furthermore, we demonstrated that some aberrant correlations that were found in the unimputed data were tempered in the imputed data.

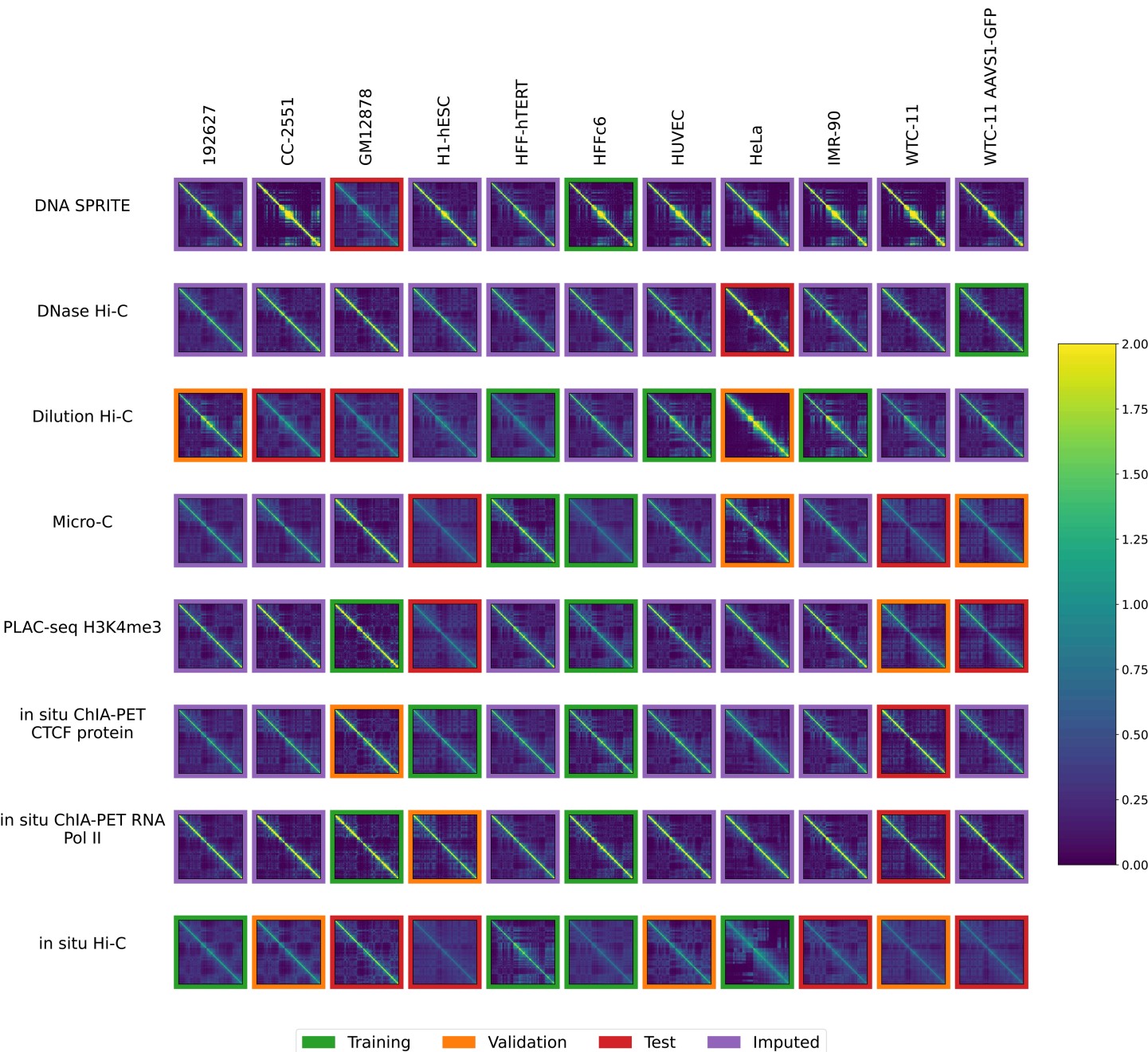

**Fig 10. Sphinx imputes contact maps for unobserved assay-cell type combinations.** The plot is similar to Fig 1, except that the unobserved contact maps (purple outline) are imputed by Sphinx. Each plot is a contact map for chromosome 1, where the number of normalized log counts is displayed as a color.

A limitation of this work is that the number of available experiments is currently somewhat small, limiting both the training and evaluation of such models. We found that increasing the number of training experiments was associated with a decrease in the MSE (Fig 3). However, we anticipate that as the cost of sequencing continues to decrease the number of available experiments, and their quality, will increase over time. In parallel, we expect that the

number of biosamples of interest will continue to increase as methods for isolating cells from conditions of interest become more sophisticated. As such, we anticipate that methods such as Sphinx will become increasingly valuable. As data becomes more robust, we envision that it may become possible to impute interchromosomal contacts, which we did not attempt with Sphinx.

## Supporting information

**S1 File.**
(PDF)

## Author contributions

**Conceptualization:** William Stafford Noble.

**Data curation:** Alan Min, William Stafford Noble.

**Formal analysis:** Alan Min, Jacob Schreiber.

**Funding acquisition:** Jacob Schreiber, Anshul Kundaje.

**Investigation:** Alan Min, Jacob Schreiber.

**Methodology:** Alan Min, Jacob Schreiber, Anshul Kundaje, William Stafford Noble.

**Resources:** Anshul Kundaje, William Stafford Noble.

**Software:** Alan Min, Jacob Schreiber, Anshul Kundaje.

**Supervision:** William Stafford Noble.

**Validation:** Alan Min.

**Visualization:** Alan Min.

**Writing – original draft:** Alan Min, William Stafford Noble.

**Writing – review & editing:** Alan Min, Jacob Schreiber, Anshul Kundaje, William Stafford Noble.

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
