## [Decision Letter · Decision Letter 0]

14 May 2025

PONE-D-25-07551Predicting chromatin conformation contact mapsPLOS ONE

Dear Dr. Noble,

Thank you for submitting your manuscript to PLOS ONE. After careful consideration, we feel that it has merit but does not fully meet PLOS ONE’s publication criteria as it currently stands. Therefore, we invite you to submit a revised version of the manuscript that addresses the points raised during the review process.

We look forward to receiving your revised manuscript.

Kind regards,

Oyelola A. Adegboye, PhD

Academic Editor

PLOS ONE

**Journal Requirements:**

1. When submitting your revision, we need you to address these additional requirements. Please ensure that your manuscript meets PLOS ONE's style requirements, including those for file naming. The PLOS ONE style templates can be found at https://journals.plos.org/plosone/s/file?id=wjVg/PLOSOne_formatting_sample_main_body.pdf and https://journals.plos.org/plosone/s/file?id=ba62/PLOSOne_formatting_sample_title_authors_affiliations.pdf 2. We note that the grant information you provided in the ‘Funding Information’ and ‘Financial Disclosure’ sections do not match.  When you resubmit, please ensure that you provide the correct grant numbers for the awards you received for your study in the ‘Funding Information’ section. 3. Thank you for stating in your Funding Statement: This work was supported by National Institutes of Health award UM1HG011531. Please provide an amended statement that declares *all* the funding or sources of support (whether external or internal to your organization) received during this study, as detailed online in our guide for authors at http://journals.plos.org/plosone/s/submit-now.  Please also include the statement “There was no additional external funding received for this study.” in your updated Funding Statement. Please include your amended Funding Statement within your cover letter. We will change the online submission form on your behalf. 4. Thank you for uploading your study's underlying data set. Unfortunately, the repository you have noted in your Data Availability statement does not qualify as an acceptable data repository according to PLOS's standards. At this time, please upload the minimal data set necessary to replicate your study's findings to a stable, public repository (such as figshare or Dryad) and provide us with the relevant URLs, DOIs, or accession numbers that may be used to access these data. For a list of recommended repositories and additional information on PLOS standards for data deposition, please see https://journals.plos.org/plosone/s/recommended-repositories. 5. Please amend either the abstract on the online submission form (via Edit Submission) or the abstract in the manuscript so that they are identical.

Reviewers' comments:

Reviewer's Responses to Questions

**Comments to the Author**

1. Is the manuscript technically sound, and do the data support the conclusions?

Reviewer #1: Yes

Reviewer #2: Yes

2. Has the statistical analysis been performed appropriately and rigorously? 

Reviewer #1: Yes

Reviewer #2: Yes

3. Have the authors made all data underlying the findings in their manuscript fully available?

Reviewer #1: Yes

Reviewer #2: Yes

4. Is the manuscript presented in an intelligible fashion and written in standard English?

Reviewer #1: Yes

Reviewer #2: Yes

5. Review Comments to the Author

**Reviewer #1:** In this work, the authors introduce Sphinx, a model designed to impute chromatin conformation contact maps. Sphinx framework predicts missing maps based on a collection of existing datasets from the 4D Nucleome (4DN) project. The authors evaluate Sphinx using cross-validation and compare it to baseline averaging methods. The analyses focus on contact decay profiles, compartments (eigenvectors), and topologically associating domains (TADs). I will try to not be repetitive given the comments of the other 3 reviewers and the authors' responses. I have some comments and questions that could help to clarify some aspects of the text and the results:

I suggest the authors to expand the discussion on the limitations of the tool to impute loops. It may require a new layer to be included in the network. It is not necessary to do it at this point but it would be beneficial to point this out.

I would recommend a quick additional analysis to benchmark when comparing the maps. Could the authors calculate the Pearson as a function of genomic distance (for example 10.1073/pnas.161360711 - SI and 10.1093/nar/gkaa818 - fig.4 G)?

Did the authors try to impute inter-chromosomal maps? That would be interesting to comment on that result or comment as a potential improvement of the model in later versions.

**Reviewer #2:** In the present manuscript by Min et al., the authors introduce Sphinx, a computational method that uses machine learning to impute missing chromatin capture contact matrices using existing experimental data as input. The main data source for the development and testing of Sphinx are datasets from the 4DN Nucleome. Existing methods use either just DNA sequence as input or require at least some additional experimental data of epigenomic profiling to exist. In contrast, Sphinx imputes contacts by using as sole input a collection of contact maps. The authors present the data, testing methodology and their conclusions appropriately. Importantly they discuss the limitations of the tool in its current state and the amount of data available.

The available submitted PDF by the authors already contains the responses to three reviewers and manuscript modifications. I understand that this correspond to a previous review round for publication at PLoS One and that these responses should be considered, but if so this was not apparent to me from the invite or the instructions provided. The authors have addressed most of the reviewers concerns and extensibly modified the manuscript. My reviewer recommendation is a minor revision with my specific comments below:

Figure 1. The introduction states “among the 88 possible contact maps, 4DN has carried out 41. Our goal is to accurately impute the missing 47 contact maps”. However, the figure legend reads “44 non-missing contact maps”.

Figure 2. Scheme is unclear, text labels are inconsistently positioned, some boxes are empty with unclear meaning, why positions are highlighted with blue circles is not described

Page 6. MSE (mean squared error) is an undefined acronym, even though very common I think the authors should define it once when first mentioned in the text.

Figure 4 is referenced in the text out of order (after Figure 9), figure numbers should be arranged in the order they are referenced in the text

There are some missing references in reviewer responses (both to papers, sections and figures in the manuscript), probably errors in the LaTeX formatting

6. PLOS authors have the option to publish the peer review history of their article (what does this mean?). If published, this will include your full peer review and any attached files.

Reviewer #1: No

Reviewer #2: No

---

## [Author Response · Author response to Decision Letter 1]

24 Jul 2025

Please see uploaded PDF response letter.

---

## [Decision Letter · Decision Letter 1]

12 Aug 2025

Predicting chromatin conformation contact maps

PONE-D-25-07551R1

Dear Dr. Noble,

We’re pleased to inform you that your manuscript has been judged scientifically suitable for publication and will be formally accepted for publication once it meets all outstanding technical requirements.

Kind regards,

Oyelola A. Adegboye, PhD

Academic Editor

PLOS ONE

Additional Editor Comments (optional):

Reviewers' comments:

Reviewer's Responses to Questions

**Comments to the Author**

1. If the authors have adequately addressed your comments raised in a previous round of review and you feel that this manuscript is now acceptable for publication, you may indicate that here to bypass the “Comments to the Author” section, enter your conflict of interest statement in the “Confidential to Editor” section, and submit your "Accept" recommendation.

Reviewer #1: All comments have been addressed

Reviewer #2: All comments have been addressed

2. Is the manuscript technically sound, and do the data support the conclusions?

Reviewer #1: Yes

Reviewer #2: Yes

3. Has the statistical analysis been performed appropriately and rigorously? 

Reviewer #1: Yes

Reviewer #2: Yes

4. Have the authors made all data underlying the findings in their manuscript fully available?

Reviewer #1: Yes

Reviewer #2: Yes

5. Is the manuscript presented in an intelligible fashion and written in standard English?

Reviewer #1: Yes

Reviewer #2: Yes

6. Review Comments to the Author

Reviewer #1: (No Response)

Reviewer #2: The authors have modified the manuscript and included he necessary corrections to address my raised comments and concerns as a reviewer.

7. PLOS authors have the option to publish the peer review history of their article (what does this mean?). If published, this will include your full peer review and any attached files.

Reviewer #1: No

Reviewer #2: No

---

## [Editor Report · Acceptance letter]

PONE-D-25-07551R1

PLOS ONE

Dear Dr. Noble,

I'm pleased to inform you that your manuscript has been deemed suitable for publication in PLOS ONE. Congratulations! Your manuscript is now being handed over to our production team.

Kind regards,

on behalf of

Assoc Prof Oyelola A. Adegboye

Academic Editor

PLOS ONE